# Antiparasitic Activity of Isolated Fractions from *Parthenium incanum* Kunth against the Hemoflagellate Protozoan *Trypanosoma cruzi*

**DOI:** 10.3390/antibiotics13070622

**Published:** 2024-07-04

**Authors:** David Alejandro Hernández-Marín, Rocio Castro-Rios, Abelardo Chávez-Montes, Sandra L. Castillo-Hernández, Joel Horacio Elizondo-Luevano, Martín Humberto Muñoz-Ortega, Eduardo Sánchez-García

**Affiliations:** 1Departamento de Microbiología, Centro de Ciencias Básicas, Benemérita Universidad Autónoma de Aguascalientes, Aguascalientes 20100, AG, Mexico; david.hernandezm@edu.uaa.mx; 2Departamento de Química Analítica, Facultad de Medicina, Universidad Autónoma de Nuevo León, Monterrey 66460, NL, Mexico; rocio.castrors@uanl.edu.mx; 3Departamento de Química, Facultad de Ciencias Biológicas, Universidad Autónoma de Nuevo León, San Nicolás de los Garza 66455, NL, Mexico; abelardo.chavezmn@uanl.edu.mx; 4Departamento de Alimentos, Facultad de Ciencias Biológicas, Universidad Autónoma de Nuevo León, San Nicolás de los Garza 66455, NL, Mexico; sandra.castillohrn@uanl.edu.mx; 5Laboratorio de Ciencias Naturales, Facultad de Agronomía, Universidad Autónoma de Nuevo León, General Escobedo 66050, NL, Mexico; joel.elizondolv@uanl.edu.mx; 6Departamento de Química, Centro de Ciencias Básicas, Benemérita Universidad Autónoma de Aguascalientes, Aguascalientes 20100, AG, Mexico; humberto.munozo@edu.uaa.mx

**Keywords:** *Parthenium incanum*, trypanocidal, parthenin, coronopolin, FT-IR, HPLC–MS

## Abstract

This study focused on isolating, identifying, and evaluating the trypanocidal potential against the hemoflagellate protozoan *Trypanosoma cruzi* of compounds from *Parthenium incanum* (Mariola), a plant used in traditional Mexican medicine to treat stomach and liver disorders. *P. incanum* has a wide distribution in Mexico. This study found that methanolic extracts of *P. incanum*, obtained by static maceration and successive reflux, had promising results. The fractions were compared using thin-layer chromatography (TLC) and those that showed similarities were mixed. A bioguided assay was performed with *Staphylococcus aureus* ATCC 25923, using agar diffusion and bioautography techniques to determine the preliminary biological activity. The fractions with antimicrobial activity were purified using a preparative thin-layer chromatography (PTLC) plate, obtaining the bioactive bandages that were subjected to a trypanocidal evaluation against the Ninoa strain of *T. cruzi* in its epimastigote stage. This revealed an IC_50_ of up to 45 ± 2.5 µg/mL, in contrast to the values obtained from the crude extracts of less than 100 µg/mL. The TLC, Fourier-transform infrared spectroscopy (FT-IR), and high-performance liquid chromatography coupled with mass spectrometry (HPLC–MS) techniques were used to identify the compounds, demonstrating the presence of sesquiterpene lactones, parthenin, and coronopolin. We concluded that these compounds have the potential to inhibit *T. cruzi* growth.

## 1. Introduction

The genus *Parthenium* contains approximately 16 species of shrubs, herbaceous perennials, and annuals. *Parthenium incanum* (also called Mariola) is a plant used in traditional Mexican medicine to treat gastric disorders, such as constipation, diarrhea, poor digestion, and stomach pain, and to treat liver conditions [1]. In addition, recent reports have attributed its antinociceptive activity to a compound called argentatin C [2]. However, very few scientific studies have studied this plant’s medicinal properties [2]. In contrast, there are a significant number of reports of harmful effects in humans and livestock caused by *P. hysterophorus*, including severe contact dermatitis and allergic respiratory problems, among other conditions [3]. It is also known to contain toxic bitter compounds called sesquiterpene lactones, which have various biological activities, including antimicrobial activity against Gram-positive and Gram-negative bacteria [4] and antiparasitic activity against trematode worms of the genera *Schistosoma* [5], *Leishmania,* and *Trypanosoma* [6]. Furthermore, studies on *Parthenium hysterophorus* have demonstrated its positive biological activity as an analgesic, anti-inflammatory, antimicrobial, and antimalarial [3,7]. In the case of *P. incanum*, hemolysis in erythrocytes and toxicity in brine shrimp were evaluated, showing that the plant extracts produced favorable results if they were subjected to a Soxhlet-type degreasing with non-polar solvents [8]. Because the genus *Parthenium* has a great variety of biological activities, the search for antiparasitic metabolites of *P. incanum* against blood parasites like *Trypanosoma cruzi* is worthwhile. Chagas disease, caused by the hemoflagellate parasite *T. cruzi*, is endemic in Mexico, Central America, and South America and affects more than 6 million people. Its limited treatment alternatives (benznidazole and nifurtimox), as well as its adverse side effects, necessitate a search for new alternatives to treat this disease [9,10]. Diverse methods for isolating and identifying compounds with biological activity have been widely reported [11,12]. The procedure for obtaining these compounds is based on a series of steps. The first crucial step in investigating new compounds is determining the type of solvent to be used in the extraction [11], depending on the compound sought [8,12]. Likewise, a phytochemical screening assay must be performed; this is a quick, simple, and inexpensive procedure that provides results on the different phytochemicals of whole plant extracts or fractions. Subsequently, different techniques, such as column chromatography (CC), thin-layer chromatography (TLC), bioautographic methods, high-performance liquid chromatography (HPLC), high-performance liquid chromatography coupled with mass spectrometry (HPLC–MS), and Fourier-transform infrared spectroscopy (FT-IR), can be used for the separation, isolation, and identification of active compounds [8,11,12,13,14,15].

This study aimed to determine whether fractions obtained from *P. incanum* extracts have trypanocidal potential using directed chromatographic and spectroscopic techniques.

## 2. Results

### 2.1. Extraction of Plant Material

The methanolic extracts were obtained using two different extraction techniques: static maceration and the Soxhlet extraction method. The yield of the maceration technique was 13.5% while that of the Soxhlet extraction method was 7.2%.

### 2.2. Preliminary Phytochemical Screening of the Methanolic Extracts

The qualitative phytochemical analysis showed several families of secondary metabolites for both methanolic extracts, including the presence of alkaloids, flavonoids, sesquiterpene lactones, sterols, triterpenes, and tannins (Table 1).

### 2.3. Chromatographic Separation

After phytochemical analysis, the methanolic extracts were fractionated using column chromatography and 40 fractions of each extract were obtained. Stepwise elutions were collected separately, concentrated, and subjected to TLC. All collected fractions showing a similar TLC pattern were pooled and tested for antimicrobial activity.

### 2.4. Bioguided Assay

According to the disc-diffusion method and bioautographic results, the antimicrobial activity was evidenced in five fractions obtained from the maceration extract and three fractions from the Soxhlet extraction, all from the elution using chloroform/methanol ratios of 9.5:0.5 and 9:1 at the chromatographic column. Subsequently, the purity of the compounds was evaluated through TLC, as previously described; equal fractions were mixed to increase their concentration, resulting in two active fractions, named “maceration-1” and “Soxhlet-1”. The weight of the active fractions was 90 mg and 105 mg, respectively. Phytochemical screening was repeated on the fractions, resulting in a strong positive reaction to sesquiterpene lactones. Each fraction was subjected to preparative thin-layer chromatography (PTLC) to recover the biologically active compound.

### 2.5. Analysis of Active Fractions

FT-IR spectra were obtained for the active fractions, maceration-1 and Soxhlet-1. As can be observed in Figure 1a, the IR spectrum for the maceration-1 fraction suggested the presence of the following functional groups: hydroxyl, OH, (3396 cm^−1^), C-H (stretching alkanes, 2294 cm^−1^ and 2854 cm^−1^) for C-H (stretching alkane), 1707 cm^−1^ for the carbonyl group (C=O), 1458 cm^−1^ for C=C (double bond), 1377 cm^−1^ for C-H (stretching alkane), and 1104 ± 5 cm^−1^ for C-O (lactone ring).

A similar spectrum was obtained for the Soxhlet-1 fraction (Figure 1b), as can be observed with signals indicating the presence of the following hydroxyl groups: 3327 cm^−1^ (OH), 2942 and 2831 cm^−1^ (CH_2_), 1741 cm^−1^ (C=O), 1449 cm^−1^ (C=C), 1337 cm^−1^ (CH3), and 1021 cm^−1^ (C-O).

UHPLC–MS was used to analyze the Maceration-1 and Soxhlet-1 fractions. Figure 2 showcases the Total Ion Chromatograms for these extract fractions. Peak signals alongside their MS and MS/MS spectra were studied. Peaks at 26.7 and 28.05 min for both fractions displayed the most prevalent ions at *m*/*z* 263 and 265. The MS/MS spectra indicated the presence of parthenin and coronopolin. Figure 3 illustrates the ESI-MS/MS spectra and Table 2 summarizes the observed fragment ions.

Chromatographic signals at 26.7 and 28.05 min in the maceration-1 and Soxhlet-1 fractions showed ions at *m*/*z* 263 and 265, suggesting the presence of parthenin and coronopolin. MS/MS was used for this ion analysis to confirm this supposition. Figure 3 shows the obtained MS/MS spectra and Table 2 summarizes the observed ions.

### 2.6. Identification of Compounds Present in the Active Fractions

The identity of the compounds present in the active fraction, maceration-1 and Soxhlet-1, was confirmed using electrospray MS/MS analysis in the positive mode for both ions (Figure 4).

### 2.7. Evaluation of the Trypanocidal Activity

The crude extracts were first evaluated for trypanocidal activity and showed an IC_50_ of less than 100 µg/mL. The extracts obtained by maceration had an IC_50_ of 54.63 µg/mL ± 3.5 µg/mL while the IC_50_ of the extract obtained by Soxhlet extraction was 63.09 µg/mL ± 4.3 µg/mL. On the other hand, the isolated fractions obtained from maceration and Soxhlet extraction presented activity at lower concentrations, with an IC_50_ of 50.5 µg/mL ±3 µg/mL and 45.0 ± 2.5 µg/mL, respectively.

## 3. Discussion

This study aimed to determine whether the fractions obtained from *P. incanum* extracts have trypanocidal potential. This might be relevant to the treatment of Chagas disease, caused by the hemoflagellate parasite *T. cruzi*, which is endemic in Mexico, Central America, and South America and affects more than 6 million people.

The presence of *P. incanum* has been recorded in many different states of Mexico, such as Aguascalientes (municipality of Asientos, Tepezalá, and the community of Juan El Grande), Chihuahua, Coahuila, Durango, Hidalgo, Guanajuato, State of Mexico, Nuevo León, Michoacán, San Luis Potosí, Querétaro, Sonora, Tamaulipas, and Zacatecas, with [7,16].

Methanolic extractions were selected for the search for metabolites with antimicrobial activity. Methanol is commonly used for the extraction of polar bioactive components since it can dissolve and extract most of the active compounds of interest [11,17]. The difference between the extraction yields may have been due to successive extractions using the Soxhlet apparatus since non-polar compounds were previously eliminated from the methanolic extracts with hexane and chloroform [8]. This did not occur during the maceration extraction since only a single solvent was used. Notably, the yield of the total Soxhlet extraction (hexane + chloroform + methanol) was 12.2%. However, when we compare the extraction yields with those obtained from other species of *Parthenium*, Bashar et al. (2022) [18] and Motmainna et al. (2021) [19] reported a higher extraction yield than ours of 18.56% and 17.56%, respectively. In these cases, *P. hysterophorus* species were used and the extraction was performed using 80% methanol. The observed difference between *P. hysterophorus* and *P. incanum* may have been affected by the type of extraction. Additionally, Frey et al. (2024) [20] highlighted that various sesquiterpene lactones produced by different plant species of the *Parthenium* genus are affected by climatic conditions and the abiotic factors that surround them during their development. Although the synthesis of sesquiterpene lactones is not considered season-dependent, some reports indicate that the synthesis of these compounds increases significantly during autumn [21].

Qualitative phytochemical analyses of both methanolic extracts from *P. incanum* showed similar results to two previously reported methanolic extracts from *P. hysterophorus* leaves [22,23]. Fractionation using column chromatography and their subsequent comparison of the fractions obtained using TLC allowed us to obtain some specific regions and the active fractions of the plant under study [24]. Column chromatography techniques have been widely used as purification techniques for the successful fractionation and isolation of desired bioactive compounds from complex extract material due to their simplicity, convenience, specificity, and availability for use in multiple mobile phases with different polarities. They are used to isolate and purify the active compounds that are responsible for the bioactivities, such as antimicrobial, antioxidant, or cytotoxic activities [25,26].

Following the previously mentioned processes and according to the results of the active fractions identified using disc-diffusion and bioautographic methods, we carried out the detection of parthenin by eluting the compounds using TLC with a mixture of chloroform/acetone (3:1). When sprayed with the vanillin reagent and sulfuric acid, they showed a violet-blue band with an Rf value of around 0.6 after heating at 70 °C for 5 min, which is characteristic of parthenin [27]. Subsequently, the fraction was recovered using PTLC with the help of a vanillin reagent. This was carried out because this methodology can be used to isolate some compounds of interest, as in the case of isolated antimicrobial compounds of *Dodonaea viscosa* [28] obtained using PTLC. The high antimicrobial potential of parthenin has been previously demonstrated [3,29]; because of this, antimicrobial activity was used as a biological indicator for the isolation of parthenin.

The FT-IR spectra obtained for the two fractions, presented in Figure 1, showed signals that indicated the presence of a lactone ring [30,31]. Figure 2 presents the chromatograms generated using UHPLC–MS for the maceration-1 and Soxhlet-1 fractions. Both fractions showed several peaks, with two significant signals emerging: one with a retention time of 26.7 min and a base peak at *m*/*z* 263 and another eluting at 28 min with a base peak at *m*/*z* 265. These peaks were likely due to the protonated molecular ion [M + H]^+^ of parthenin (molecular weight: 262 g/mol) and coronopolin (molecular weight: 264 g/mol). Figure 4 illustrates the chemical structures of these sesquiterpene lactones.

To verify the identity of these compounds, a MS/MS analysis was carried out in the positive mode for both ions [32]. Figure 3 shows the resulting spectra and Table 2 summarizes the fragmentation data for the ions at *m*/*z* 263 and 265 detected in the active fractions of *P. incanum* Kunth. As noted earlier, these ions at *m*/*z* 263 and 265 could correspond to the protonated molecular ions [M + H]^+^ of the pseudoguaianolide sesquiterpene lactones, specifically parthenin and coronopolin.

A characteristic loss of water, [M + H − 18]^+^, was observed for both compounds, leading to fragments at *m*/*z* 245 for parthenin and 247 for coronopolin. Product ions at *m*/*z* 227 and 229 for parthenin and coronopolin, respectively, were explained as a result of a C=O loss, yielding the [M + H − 36]^+^ ion. Fragments at *m*/*z* 181 and 183 (from transitions 209 → 181 and 211 → 183) could be attributed to a characteristic loss of CH_2_=CH_2_. dditional product ions formed during the MS/MS experiments included 209 [M + H − 54]^+^, 199 [M + H − 64]^+^, and 181 [M + H − 82]^+^ for parthenin and 219 [M + H − 46]^+^, 205 [M + H − 60]^+^, 201 [M + H − 64]^+^, 187 [M + H − 78]^+^, 183 [M + H − 92]^+^, and 147 [M + H − 118]^+^ for coronopolin.

Despite the lack of Nuclear Magnetic Resonance (NMR) analysis, the evidence supporting the presence of parthenin and coronopolin in plants from the genus *Parthenium*, combined with the agreement of MS/MS fragmentation patterns with the existing literature, allows for the confident assignment of the chromatographic signals at 26.7 and 28 min to parthenin and coronopolin, respectively [27,33,34,35,36,37].

The assays conducted in this investigation were performed in vitro on the epimastigote stage because, in a recent article published by our research team, a positive correlation between trypanocidal activity against epimastigotes in vitro and trypomastigotes in vivo was established. Thus, the activity of the trypomastigote stage of a compound or metabolite can be estimated using the in vitro epimastigote assay [38,39,40]; moreover, this in vitro evaluation is simpler, more accessible, and faster in terms of obtaining results.

The data obtained from the trypanocidal activity suggested that the isolated fractions had a slightly lower concentration to inhibit *T. cruzi* than the crude extracts (concentrations close to 50 µg/mL). These concentrations were considered representative of moderate activity (IC_50_: 45 ± 2.5 µg/mL) to fair activity (IC_50_: 50.5 µg/mL ± 3.0 µg/mL), according to the classification proposed by Ohashi et al. (2018) [39]. Our results were superior to those obtained by Muñoz et al. (2012), who determined, in vitro, the effect of the antiparasitic Nifurtimox and Benznidazole; those results showed an IC_50_ of 2.34 µg/mL ± 0.72 µg/mL and 13.12 µg/mL ± 2.45 µg/mL, respectively [41]. Likewise, the data obtained by Acosta et al. (2020), in which the same antiparasitic agents were evaluated against several *T. cruzi* clones, showed a maximum IC_50_ of 15.22 µg/mL ± 3.10 µg/mL [9].

Our results indicated that the trypanocidal activity of the isolated fractions was weak when compared in the first instance with Nifurtimox and Benznidazole (approved and recognized by PAHO/WHO). However, these drugs have sales restrictions, their use and distribution are regulated, they cause adverse effects, and they must be used in certain phases of the disease to be effective [10,42]. Despite their low activity compared with synthetic compounds, natural products have made an important contribution to antiparasitic drug research. Likewise, the isolated fractions of parthenin and coronopolin presented a higher degree of activity against the epimastigote stage than vestitiol (5.2 µg/mL), an isoflavonoid isolated from a lyophilized Red Propolis [43]. Furthermore, in a 2021 study (Bethencourt-Estrella et al.) of 13 compounds isolated from the zoanthid *Palythoa aff. clavata*, three of the best results against the *T. cruzi* epimastigote were obtained for sesquiterpene lactones, i.e., anhydroartemorin, cis- and trans-costunolide-14-acetate, and 4-hydroxyarbusculin A [44]. However, it is important to note that it is difficult to compare the activity of pure synthetic compounds with fractions isolated from natural sources; in this case, parthenin and coronopolin should be purified and evaluated individually to compare their activity with isolated compounds.

The present investigation demonstrated that the compounds present in isolated fractions of *P. incanum* have trypanocidal activity. In addition, it showed that the bioactive compounds, even from different extractions, maintained their biological activities.

## 4. Materials and Methods

### 4.1. Plant Material

The aerial parts (stems and leaves) of *P. incanum* were collected from the wild in August of 2021 in the municipality of Real de Asientos, Aguascalientes, Mexico (22°13′47.2″ N, 102°06′20.9″ W). A voucher sample of the plant was deposited at the herbarium of Facultad de Ciencias Biológicas of Universidad Autónoma de Nuevo León for taxonomic identification by Dr. Marco Antonio Guzmán Lucio.

### 4.2. Extraction

Two different extraction methods were carried out: static maceration and reflux by the Soxhlet apparatus. For maceration, 100 g of dried and powdered plant material was extracted with 500 mL of methanol (CTR, Scientific, Monterrey, Mexico) for 24 h at 25 °C without shaking; the samples were then kept in dark conditions. After the maceration time, the solvent was filtered using Whatman No. 1 filter paper. After that, it was completely evaporated using a rotary evaporator (Yamato Scientific Co., Ltd., RE 200, Tokyo, Japan). Finally, the extract was resuspended in 10 mL of the extraction solvent, mixed, and completely evaporated for dry weight determination. To obtain the extract using reflux (Soxhlet, Pyrex, Corning, NY, USA) 200 g of the dried and crushed plant was heated at 45 °C to reflux for 48 h with 500 mL of hexane (CTR, Scientific, Monterrey, Mexico), chloroform (CTR, Scientific, Monterrey, Mexico), and methanol (CTR, Scientific, Monterrey, Mexico), separately and in sequence. Non-polar extracts were discarded, and the methanolic extract was evaporated entirely under reduced pressure to determine its dry weight. The extracts (obtained through maceration and reflux) were suspended in 15 mL of absolute methanol, aliquoted in amber vials, and stored at 4 °C until needed [8].

### 4.3. Phytochemical Screening

Qualitative phytochemical screening was carried out using specific reactions to demonstrate the presence of secondary metabolites. The presence of unsaturated groups, such as triterpenes, sterols, coumarins, alkaloids, sesquiterpene lactones, quinones, carboxyl group, tannins, flavonoids, saponins, and carbohydrates, in the extracts was proven [45].

### 4.4. Chromatographic Separation

The extracts (5 g) were fractionated through column chromatography using silica gel technical grade (Ca ~0.1%) 60 Å of 200–400 mesh (Sigma Aldrich, St Louis, MO, USA). The extracts were eluted using 2 L of different combinations of chloroform (CTR, Scientific) and methanol (CTR, Scientific), with increasing polarity. Fractions were collected in 50 mL volumes. All the obtained fractions were concentrated in a rotary evaporator (Yamato Scientific Co., Ltd., RE 200). Once concentrated, the dried fractions were resuspended in 10 mL of the appropriate solvent and placed in 25 mL beakers. Similarities between fractions were evaluated using TLC, with aluminum foils having silica gel 60 F_254_ (Merck, Darmstadt, Germany). Similar fractions were combined [24].

### 4.5. Bioguided Assay: Detection, Isolation, and Identification of Active Compounds

Fractions with antimicrobial activity were selected using the disc-diffusion method, with 6 mm filter paper discs impregnated with approximately 5 mg of each fraction and placed on Mueller–Hinton agar plates, previously seeded with *Staphylococcus aureus* ATCC 25923 (1 × 10^6^ CFU/mL). After the incubation period (overnight), antimicrobial activity was detected based on an inhibition zone surrounding the disc [46,47,48]. The detection of antibacterial compounds present in the selected active fractions was made using the agar overlay bioautography method. For this, 10 cm × 20 cm pre-activated TLC plates (silica gel type G; 5–15 µm F_254_; Sigma-Aldrich, St Louis MO, USA) were used as the stationary phase. TLC plates were loaded with 10 µL of each selected active fraction in a narrow band; after this, the plates were developed in a solvent chamber using chloroform/acetone (3:1) as a mobile solvent system. The developed plates were air-dried overnight at 35 °C to remove the residual solvent. After this, the chromatogram was covered with molten MH Agar (Difco, Detroit, MI, USA) and maintained in a water bath at 45 °C previously seeded with a suspension of *S. aureus* 25923 at a final concentration of 1 × 10^8^ CFU/mL. After solidification, the plates were incubated overnight at 37 °C in a humidity chamber. The active fraction was visualized through the appearance of clear zones after the incubation period [27,48,49]. The Rf value of the chromatogram was calculated and recorded.

### 4.6. Preparative Thin-Layer Chromatography (PTLC)

Once the antimicrobial band was detected through bioautography, the active fraction was purified through PTLC. This was performed on 20 cm × 20 cm glass plates, precoated (1 mm) with silica gel 60 G (Sigma-Aldrich, St Louis, MO, USA). The active fraction (50 µL) was applied as a narrow band, about 1.5 cm in height from the lower edge of the plates, leaving a 1 cm border on the sides of the plate. The plates were developed in a chromatographic chamber after conditioning for 20 min with mobile phase vapor, using the appropriate solvents, as previously described. After development, the plates were removed from the system and dried in a fume hood. The bands responsible for the antimicrobial activity were scraped out and recovered with methanol, according to the Rf previously obtained [28]. The compounds were placed in amber vials and were completely dried for further analysis. Once the antimicrobial compounds were obtained, phytochemical screening was performed, as previously mentioned, to determine the identity of the isolated fractions.

### 4.7. Identification of Isolated Compounds Using FT-IR and UHPLC–MS Analyses

An amount of 2 mg of each isolated compound was used for the FT-IR analysis [50]. The FT-IR analysis was carried out using a Perkin Elmer Spectrum GX, with 16 scans in the region of 4000–650 cm^−1^. Chemical tests were carried out to confirm the type of secondary metabolite [28,29]. For the analysis of UHPLC–MS, chromatographic separation was performed using an Ultra HPLC Ultimate 3000 chromatographic system consisting of a high-pressure gradient pump with an online degasser, an autosampler, a column oven, and a variable wavelength UV–Vis detector (Thermo Fisher Scientific Dionex, Waltham, MA, USA). A Discovery HS F5 column (150 × 2.1 mm, 3 μm; Supelco, Bellefonte, PA, USA) was used with a mixture of a 2% (*v*/*v*) acetic acid aqueous solution and acetonitrile as a mobile phase. Gradient elution started with a 5-min period with 5% acetonitrile and increased to 50% in 45 min. After this, a 10-min washing step with 80% acetonitrile was included and, then, the mobile phase was returned to the initial composition, with conditioning for 20 min before the next injection. The mobile phase flow was 200 µL/minute, the column temperature was 50 °C, and the injection volume was 10 µL. Detection was performed with mass spectrometry using an LCQ Fleet (Thermo Fisher Scientific, Waltham, MA, USA) equipped with an electrospray ionization source (ESI) and an ion trap mass analyzer. Nitrogen was used as a sheath and drying gas at a flow of 40 and 10 units, respectively, and the analyses were performed using the following parameters: spray voltage 5 kV, capillary voltage 10 V, capillary temperature 325 °C, and lens tube voltage 60 V. Data acquisition was performed in the positive mode and included a full scan from *m*/*z* 100 to 1000 and mass/mass experiments using the collision-induced dissociation (CID), with a normalized collision energy of 24.5%, an isolation width of 0.9 *m*/*z*, an activation Rf voltage (activation Q) of 0.25, an activation time of 30 ms, and a scanning range of *m*/*z* 100 to 300. Samples were dissolved using methanol (LC–MS grade) and filtered through a 0.2 µm nylon membrane (Millipore, Burlington, MA, USA).

### 4.8. Evaluation of the Trypanocidal Activity of Extracts and Isolated Fractions

The trypanocidal effect of the crude extracts and isolated fractions on the Ninoa strain of *T. cruzi* was determined using a microdilution method. For this, 140 µL of epimastigotes (1 × 10^5^ /mL) growing in the Novy–MacNeal–Nicolle (NNN) culture medium was placed in each well. Afterwards, 10 µL of the compound at different concentrations (1000–10 µg/mL) were added. The negative control was epimastigotes without treatment; 1% crystal violet was used as the positive control. Methanol was used as a blank; it was the vehicle of the compound to be evaluated. The assay was performed two times in triplicate. The plates were incubated for 24 h at 25 ± 1 °C in a humid chamber. Viability was determined by microscopic observations using a Neubauer chamber [51]. The inhibitory concentration 50 (IC_50_) was obtained using the Probit method with the SPSS software ver. 23 [52].

## 5. Conclusions

This study confirmed the trypanocidal potential of two fractions of *P. incanum* against *T. cruzi*. In future research, it will be important to purify the compounds present in the active fraction and evaluate the individual activities of parthenin and coronopolin. Moreover, toxic and cytotoxic evaluations of parthenin and other bioactive compounds of the *Parthenium* genus should be performed to decipher their possible therapeutic uses.

## Figures and Tables

**Figure 1 antibiotics-13-00622-f001:**
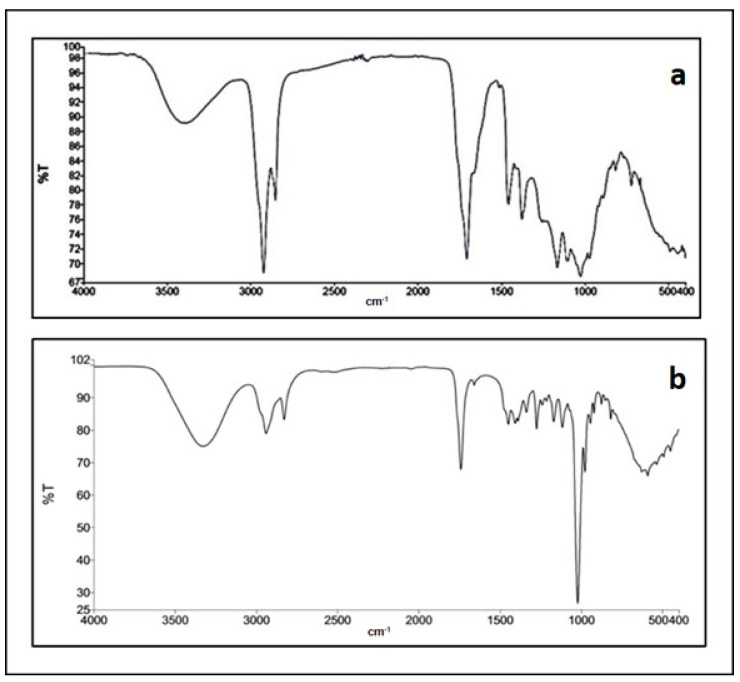
FT-IR spectra obtained for the (**a**) maceration-1 and (**b**) Soxhlet-1 active fractions of *Parthenium incanum* Kunth.

**Figure 2 antibiotics-13-00622-f002:**
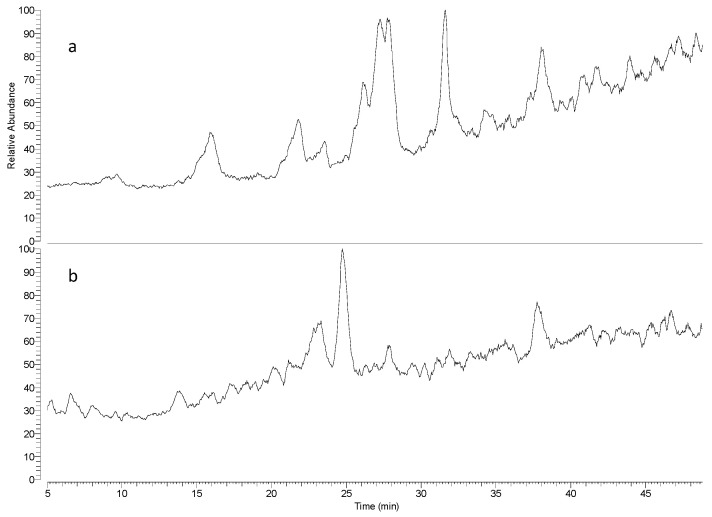
Total Ion Chromatograms obtained for the (**a**) maceration-1 and (**b**) Soxhlet-1 isolated active fractions.

**Figure 3 antibiotics-13-00622-f003:**
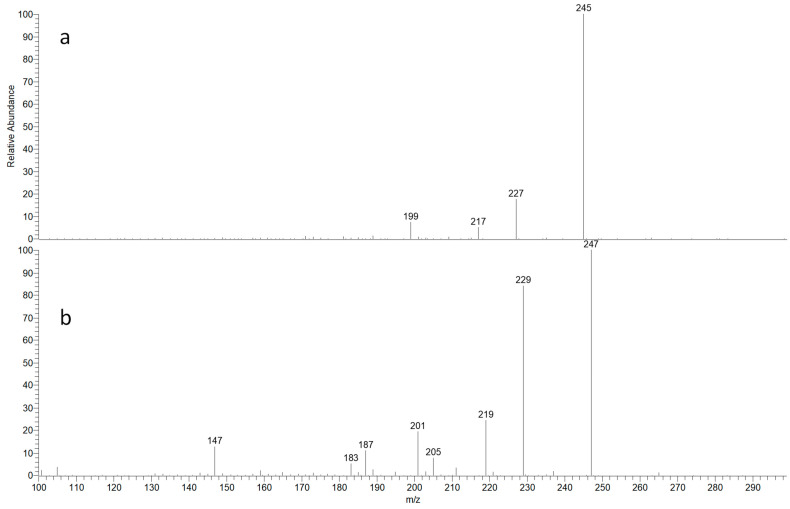
MS/MS spectra using, as precursors, the ions at (**a**) *m*/*z* 263 and (**b**) *m*/*z* 265.

**Figure 4 antibiotics-13-00622-f004:**
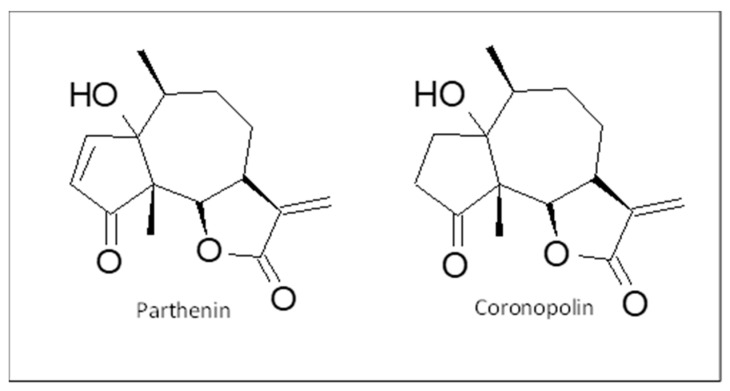
Chemical structures of parthenin and coronopolin.

**Table 1 antibiotics-13-00622-t001:** Preliminary phytochemical screening of methanolic extracts.

Test	Maceration	Soxhlet
Flavonoids	+	+
Alkaloids	−	+
Tannins	+	+
Sesquiterpene lactones	+	+
Coumarins	+	+
Carbohydrates	+	+
Sterols	−	+
Terpenes	−	+
Saponins	−	−
Quinones	+	−
Carboxyl groups	−	−

**Table 2 antibiotics-13-00622-t002:** Compound identification using MS/MS in *Parthenium incanum* Kunth active fractions.

Compound	Precursor Ion (*m/z*)	Product Ions (*m/z*)
Parthenin	263	245, 227, 217, 209, 199, 181
Coronopilin	265	247, 229, 219, 211, 205, 201,187, 183, 147

## Data Availability

The data used to support the findings of this study are available from the corresponding author upon request.

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
