# Peer review of "Antiparasitic Activity of Isolated Fractions from Parthenium incanum Kunth against the Hemoflagellate Protozoan Trypanosoma cruzi"

_antibiotics, 2024, doi:10.3390/antibiotics13070622_

Round 1

Reviewer 1 Report

Comments and Suggestions for Authors

Comments to the Authors

The manuscript entitled “Trypanocidal activity of compounds isolated from Parthenium incanum Kunth, against the hemoflagellate protozoan Trypanosoma cruzi” has been reviewed. The topic is quite innovative and interesting, the MS is acceptable in the present form.

Additionally:

Line 85 – Table 1 has two numbers superimposed on the letters at the end, please check.

Line 166 – Figure 7 has two numbers superimposed on the molecule structure, please check.

Lines 383-384 - Parthenium incanum should be in Italic.

Reviewer 2 Report

Comments and Suggestions for Authors

This study is about determining the trypanocidal activity of compounds isolated from Parthenium incanum against Trypanosoma cruzi. The work is beautifully constructed. But please use more details about the identification of the plant, especially in the method section. Add the plant part used. The findings are quite mixed. Be simpler and more descriptive. Explain the figures in more detail. Explain what the values mean. There are studies about this plant in the literature. Mention these studies in the introduction section. The article can be published after revision.

Reviewer 3 Report

Comments and Suggestions for Authors

The manuscript describes the purification of two sesquiterpene lactones from Parthenium incanum and the anti-Trypanosoma cruzi activity of one of these molecules. However, it is already known that sesquiterpene lactones exhibit anti-Trypanosoma cruzi activity. Similarly, the presence of parthenin in Parthenium extracts has been previously published. Thus, the novelty of the findings is compromised.

To enhance the manuscript, the authors should consider the following suggestions:

  • Compare the extraction process and yield with those of other Parthenium species. Additionally, investigate whether the production of sesquiterpene lactones is season-dependent.
  • Test the anti-Trypanosoma activity on trypomastigote forms and present the cytotoxicity data to compare the effects on both cell types.
  • Express concentrations in micrograms per milliliter (µg/mL) instead of parts per million (ppm).
  • Explain why coronopolin was not tested.
  • Expand the discussion section, which is currently only one paragraph.
  • Improve the legends of the figures. Note that Figure 4 is not cited in the text and its results are not described.

For the reasons described above, I hesitate to recommend this paper to be accepted for publication in this journal at least in its present form.

Comments on the Quality of English Language

The manuscript at many points does not read fluently. For this reason, this paper should be revised with careful attention prior to resubmission.

Reviewer 4 Report

Comments and Suggestions for Authors

The topic of this manuscript is interesting. However, the reviewer does not support the acceptance of this manuscript:

(1) For structure determination, NMR data is crucial. The identification of such compounds is incomplete.

(2) The potency is fairly weak. It is unlikely for such phytochemical to be translated into clinical setting.

(3) What is their mechanism of action?

Round 2

Reviewer 3 Report

Comments and Suggestions for Authors

The revised version of the manuscript entitled "Trypanocidal activity of compounds isolated from Parthenium incanum Kunth, against the hemoflagellate protozoan Trypanosoma cruzi" does not address most of the questions raised by this reviewer. Specifically, the response to Comment 1 needs to be incorporated into the text, and the response to Comment 2, along with the reference to the same effect on epimastigote and trypomastigote forms, must also be included. Additionally, the authors claim in their responses to Comments 4, 5, and 6 that the Results and Discussion sections were rewritten and improved; however, this does not appear to be the case. This raises the question of whether the correct revised version was uploaded. Finally, the requested improvements to the English language in Comment 7 have not been made. Therefore, I do not recommend this manuscript for publication in its current form.

Comments on the Quality of English Language

The requested improvements to the English language in Comment 7 have not been made. 

Reviewer 4 Report

Comments and Suggestions for Authors

The authors did not address the reviewer's question. Where is the NMR data? 

Round 3

Reviewer 3 Report

Comments and Suggestions for Authors

The third version of this manuscript is relevantly improved when compared to the previous one. However, the authors must check the numbers of references in the manuscript. For instance, the references cited in lines 235-240 (46-47) do not correlate to the information described; in this paragraph, reference 49 must be cited, among others. In addition, Acosta et al is cited in line 250 as reference 50, although in reference list it is number 9. This must have happened due to the inclusion of new references but it is mandatory to be corrected.

Author Response

We carefully reviewed each reference to address any inconsistencies identified by the reviewer. After this review, we are confident that each reference accurately reflects the information in the text. We also improved the references by adding DOI numbers (if available) and aligning them with the APA format.

Reviewer 4 Report

Comments and Suggestions for Authors

The manuscript has been improved. The reviewer has no objection to accept it. 

Author Response

Reviewer 4 did not request modifications to the document.

Reviewer 4's comment was as follows:

"The manuscript has been improved. The reviewer has no objection to accept it"